# Defining an Optimal Range of Centrifugation Parameters for Canine Semen Processing

**DOI:** 10.3390/ani13081421

**Published:** 2023-04-21

**Authors:** Nicole Sugai, Stephen Werre, Julie Cecere, Orsolya Balogh

**Affiliations:** 1Department of Small Animal Clinical Sciences, Virginia-Maryland College of Veterinary Medicine, Blacksburg, VA 24061, USA; 2Department of Population Health Sciences, Virginia-Maryland College of Veterinary Medicine, Blacksburg, VA 24061, USA

**Keywords:** dog, centrifugation, chilled, cooled, sperm, viability, morphology, motility

## Abstract

**Simple Summary:**

Processing canine semen is necessary for the preparation of cooled shipments or cryopreservation. Our goal was to define an acceptable range of centrifugation parameters (gravitational (g) force and time) without a severe negative impact on semen quality. We hypothesized that higher g forces (900 g vs. 400 g or 720 g) and longer treatment (10 min. vs. 5 min) improve sperm recovery rates yet causes a larger magnitude of decline in semen parameters over a 48 h cooling period. Initial raw semen evaluations served as each dog’s own control. Sperm recovery rates post-centrifugation were similar between treatment groups. Sperm plasma membrane integrity, motility and morphology were not different between treatment groups but declined over time. In conclusion, a range of 400–900 g for 5–10 min centrifugation provides clinically viable semen quality after up to 48 h of cooled storage in dogs.

**Abstract:**

Our objective was to determine a clinically relevant range of centrifugation parameters for processing canine semen. We hypothesized that higher gravitational (g) force and longer time of centrifugation would result in improved spermatozoa recovery rate (RR) but poorer semen quality. Cooled storage under standard shipping conditions was used as a stressor to evaluate long-term treatment effects. Individual ejaculates collected from 14 healthy dogs were split into six treatment groups (400 g, 720 g, and 900 g for 5 or 10 min). Sperm RR (%) was calculated post-centrifugation, and plasma membrane integrity (%, Nucleocounter^®^ SP-100™), total and progressive motility (%, subjective and computer-assisted sperm analysis), and morphology (%, eosin-nigrosin staining) were assessed on initial raw semen (T0), post-centrifugation (T1), and 24 h (T2) and 48 h (T3) after cooling. Sperm losses were minimal, and RRs were similar across treatment groups (median >98%, *p* ≥ 0.062). Spermatozoa membrane integrity was not different between centrifugation groups at any time point (*p* ≥ 0.38) but declined significantly during cooling (T1 vs. T2/T3, *p* ≤ 0.001). Similarly, total and progressive motility did not differ across treatments but declined in all groups from T1 to T3 (*p* ≤ 0.02). In conclusion, our study showed that centrifugation within a range of 400 g–900 g for 5–10 min is appropriate for processing canine semen.

## 1. Introduction

The use of cooled and frozen semen for artificial insemination (AI) for canine breeding is vital for overcoming geographic and temporal barriers. The demand for insemination in dogs has significantly increased over the last few decades. The gain in popularity for shipping cooled canine semen is due to the relative ease, decreased expense in logistics compared to frozen semen, and the reduced strain for transporting live animals. Comparable pregnancy rates achieved by cooled and fresh semen inseminations have added to the increase in popularity [1,2]. However, the gains may be countered by other biologic factors and handling conditions, which can reduce the quality of shipped cooled semen at the time of insemination [1,2,3]. These include routine processing methods that can negatively impact the viability, motility, and fertilizing capacity of the sperm cells. Additionally, the admixture of prostatic fluid and other seminal plasma factors in raw samples produces reactive oxygen species (ROS), which are known to be harmful to spermatozoa [2,3,4,5]. Therefore, centrifugation of canine semen prior to cooled shipment or cryopreservation is routinely performed in clinical practice to remove the potentially harmful effects of prostatic fluid or other contaminants such as urine [6]. The presence of such contaminants can lead to increased ROS content, changes in osmolality and pH balance, and negatively affect semen quality. Increased ROS levels can decrease sperm motility, DNA integrity and spermatozoa membrane integrity due to lipid peroxidation and may cause the apoptosis of spermatozoa [7,8,9]. The disadvantages of centrifugation are the damage to spermatozoa, loss of spermatozoa into the supernatant, and risks of decreasing fertilizing ability due to damage from the procedure itself [4,9,10]. Therefore, it is critical to optimize this one step when processing canine semen.

To date, in dogs, only one study specifically compared the effects of different centrifugation gravitational (g) forces on sperm losses and on in vitro semen quality during cooling [10]. This study provided the foundation for optimizing centrifugation for canine semen processing [10]. In their study, Rijsselaere et al. concluded that 720 g for 5 min gave the best spermatozoa recovery rate (RR) and membrane integrity combination, but they investigated a wide range of centrifugation speeds (180 g–2880 g). Although higher speed settings of 1620 g and 2880 g allowed for better sperm pellet formation reducing total sperm losses, these treatments resulted in a significantly higher number of dead and moribund sperm cells with damaged plasma membranes [10]. This study, however, used pooled canine semen and extended it prior to centrifugation, similar to stallion andrology work [10]. Other researchers used centrifugation parameters for canine semen based on the results of Rijsselaere et al., but those studies did not specifically investigate the effects of particular centrifugation parameters on semen quality [11,12]. Both studies had different objectives assessing extender and cooling while using the findings from Rijsselaere et al. for semen preparation methodology. Some manufacturers of commercial canine semen extenders also have their own recommendations for semen processing and/or washing ranging from 100 g to 150 g for 5 min and 700 g for 10 to 15 min (Zoetis, Kalamazoo, Michigan and Minitube, Tiefenbach, Germany, respectively).

In contrast, equine semen studies have reported that an increase in g forces to 1800 g and 2400 g for a short period of time (5 min) resulted in increased sperm RR without compromising semen quality when compared to the standard protocol of 600 g for 10 min [13,14]. Prior to centrifugation, the semen was diluted to 25 million/mL with a skim milk-based extender [13], which is not routinely conducted in the dog. Many assisted reproductive techniques and protocols may not apply across species due to differences in spermatozoa characteristics and plasma membrane composition [15,16,17]. Differences can be appreciated in extender formulations that are optimized for each species and in certain conditions for handling semen in terms of reducing the amount of ROS produced and phospholipid membrane damage that can occur to spermatozoa [15,18,19]. Therefore, evaluating the individual stud dog’s semen quality trajectory based on processing parameters can help standardize the field for clinical canine andrology similar to the stallion [13,20].

Ideal centrifugation would allow for maximum sperm RR without sacrificing sperm quality. Therefore, the aim of this study was to define an acceptable range of centrifugation parameters within a narrower scale of g forces (400 g–900 g) and a duration of 5 and 10 min. This investigation would highlight whether any potential differences exist between these centrifugation settings in terms of sperm RR and sperm quality parameters in an in vitro setting. We used individual dog ejaculates instead of pooled semen with each stud dog serving as their own control. Furthermore, semen samples were not extended prior to centrifugation to mimic routine clinical handling and to exclude the potential influence of the extender on the sedimentation rate. Cooling post-processing in industry-standard shipping containers was used as a stressor to evaluate the long-term effects of each centrifugation set-up. Our hypothesis was that a higher centrifugation g force and longer processing time will lead to increased sperm RR but reduced spermatozoa membrane integrity, motility, and percent normal morphology.

## 2. Materials and Methods

### 2.1. Animals 

Fourteen healthy, mature adult dogs of medium to large breeds (mean 32.1 kg, range 23.1–61.3 kg) and between two and six years of age (mean of 3.3 years) were included in this study. The breeds included Doberman Pinscher (n = 1), Irish Water Spaniel (n = 1), Labrador Retriever (n = 3), Golden Retriever (n = 3), Standard Poodle (n = 2), Pointer (n = 2), Rottweiler (n = 1), and Weimaraner (n = 1). Males participating in the study were found to be healthy at the time of semen collection based on the general physical and reproductive exams. Only dogs with no history of reproductive disease or infertility concerns and without apparent prostatic and testicular disease were included. None of the dogs were on any prescription medication or undergoing treatments for medical conditions at the time of presentation. All dogs were receiving heartworm and flea/tick preventative medication that were considered safe for breeding animals. *Brucella canis* serology (canine *Brucella* slide agglutination test (CBSA) and *Brucella canis* Agar Gel Immunodiffusion II (AGID II) at Cornell University Animal Health Diagnostic Center) was negative in all dogs at the time of inclusion. Seven dogs had never been collected or mated naturally, and seven were proven studs. Each male was enrolled on a voluntary basis from client-owned animals in compliance with the Virginia-Maryland College of Veterinary Medicine Institutional Animal Care and Use Committee (IACUC protocol number 21–194), and all owners signed an informed consent form. 

### 2.2. Semen Collection 

Semen was collected by manual technique as described previously [21] after a minimum of 7–10 days of sexual rest. If necessary, a teaser bitch was present to improve libido, otherwise, vaginal swabs collected and frozen from estrous bitches were used for stimulation. The first and second fractions of the ejaculate were collected in a disposable plastic funnel sleeve (Minitube, Tiefenbach, Germany). Semen collection was discontinued when the third fraction appeared to exclude and minimize the amount of prostatic fluid admixture to the sample. Ejaculates were transferred to a 15 mL Falcon conical centrifugation tube (Corning, Christiansburg, VA, USA) and the sample was kept at room temperature during initial processing. Only semen samples of over 1.5 mL total volume, with a minimum 70 × 10^6^/mL spermatozoa concentration, ≥70% subjective total motility and ≥40% morphologically normal spermatozoa were included in the study.

### 2.3. Semen Evaluation

Initial semen evaluation was performed immediately after the collection of the fresh (raw) ejaculate to determine the total ejaculate volume, concentration, and the total number of spermatozoa in the ejaculate, spermatozoa plasma membrane integrity (viability), motility, and morphology. Spermatozoa concentration was determined using the Nucleocounter**^®^** SP-100™ (Chemometec, Allerod, Denmark) with SP1-cassettes according to the manufacturer’s instructions using Reagent S100 lysis buffer (Chemometec, Allerod, Denmark). The total number of spermatozoa in the full ejaculate was calculated as concentration multiplied by total ejaculate volume. Sperm plasma membrane integrity (viability) was assessed using the Nucleocounter**^®^** SP-100™ according to the manufacturer’s instructions. Membrane integrity (%) was calculated as (total sperm concentration**—**concentration of non-viable sperm)/(total sperm concentration) × 100.

Total and progressive motility (%, TM and PM, respectively) were assessed by a computer assisted sperm analysis system (CASA; Sperm Vision™, Minitube, Tiefenbach, Germany). Computer assisted (CASA) motility evaluation was conducted following previously described protocols [2,22]. Leija 20 µm disposable counting chamber slides (Minitube, Tiefenbach, Germany) were pre-warmed to 37 °C and all motility evaluations were run at 37 °C using the settings presented in Table 1. If the field was too concentrated to have an accurate reading, 3 µL of the sample was diluted with 50 µL of 37 °C warmed phosphate buffered saline (PBS) (Chemometec, Allerod, Denmark) and well mixed prior to loading into the chamber. All motility assessments were conducted with the sperm concentration between 50 and 75 × 10^6^/mL, and this dilution was kept standard for each dog’s samples for the entire evaluation window. Subjective total motility was assessed by placing 10 µL of a well-mixed semen sample on a pre-warmed (37 °C) glass slide under a coverslip and examined using a phase-contrast microscope at 100× magnification. Subjective TM (%) was assessed to the nearest 5%. 

Spermatozoa morphology was examined using eosin-nigrosin staining at 1000× magnification (oil immersion) under a phase contrast microscope by the same two investigators (NS and OB). The average of the two evaluations was used for statistical analysis. Slides were prepared with 10 µL of the semen sample and 10 µL of eosin-nigrosin stain applied to a pre-warmed slide and mixed to form a monolayer. All normal and abnormal cells were recorded by counting a total of 200 cells.

### 2.4. Centrifugation and Evaluation of Post-Centrifugation Sperm Characteristics

After the initial evaluation of the fresh (raw) semen, the ejaculate was split into six equal volume aliquots. Each aliquot was placed in a 15 mL conical Falcon tube for the remainder of processing and cooled storage. The fresh (raw) semen aliquots were centrifuged at room temperature at 400 g, 720 g, and 900 g each for 5 min. (treatment groups A–C, respectively) or for 10 min. (treatment groups D–F, respectively). The order of centrifugation was randomly assigned to each aliquot. A Sorvall ST8R centrifuge (Thermo Scientific, Waltham, MA, USA) with a swinging bucket rotor and soft acceleration/deceleration was used for all samples.

After centrifugation, the supernatant was immediately and completely removed from the sperm pellet of each aliquot. For 11 ejaculates, the supernatant from each aliquot was collected and placed in 10% neutral phosphate buffered formalin (1:1 *vol/vol*) for later evaluation of sperm losses. The resulting sperm pellet was extended and resuspended with CaniPlus Chill LT (Minitube, Tiefenbach, Germany) at room temperature to the same initial volume as the raw semen aliquot. After resuspension, total sperm concentration and membrane integrity (Nucleocounter**^®^** SP-100™), motility (CASA and subjective evaluation) and morphology (eosin-nigrosin) were determined immediately after extension as described for the initial semen evaluation. The sperm recovery rate (RR1, %) was calculated for each aliquot as (sperm concentration post-centrifugation and resuspension)/(sperm concentration in raw semen) × 100, using the Nucleocounter**^®^**. Additionally, for the 11 samples for which the supernatant was collected after centrifugation, RR was also established from the recovered supernatant (RR2, %). The concentration of spermatozoa in the supernatant was determined with a hemocytometer, and RR2 (%) was calculated as (sperm concentration in raw semen—sperm concentration in supernatant)/(sperm concentration in raw semen) × 100.

### 2.5. Cooling and Evaluation of Sperm Characteristics at 24 h and 48 h after Cooling 

To better assess the long-term effects of each centrifugation setting on sperm quality, cooling was used as a stressor. Following semen evaluation post-centrifugation and extension, the extended semen aliquots were packaged in a canine semen transport shipping box (Minitube, Tiefenbach, Germany) for 48 h at 4 °C. The Falcon tubes were closed, wrapped with Parafilm**^®^** (Sigma-Aldrich, St. Louis, MO, USA) to secure the lid, and packaged in a Ziplock bag for secondary containment. The two aliquots of the same centrifugation gravitational (g) force setting per dog were then wrapped in a paper towel and placed in the center portion of the shipping box with ice packs in the correct position on the sides. The samples underwent a curvilinear cooling rate to reach approximately 4 °C for storage duration, with the storage boxes kept in an ambient environment of about 21 °C [23]. The ice packs for each box were replaced at 24 h at the time of semen evaluations. Spermatozoa plasma membrane integrity, TM and PM (subjective and computer assisted (CASA)), and morphology were determined at 24 h and 48 h of cooled storage. Cooled samples were allowed to equilibrate at 37 °C for 2–5 min before motility evaluations.

### 2.6. Statistical Analysis

To adjust for differences in individual ejaculate characteristics, parameters of the initial fresh (raw) semen evaluation at time 0 (T0) were used as a baseline for subsequent semen evaluations for each dog. This allowed the individual dog to stand as his own control for reference on potential differences between centrifugation treatment groups and changes over time.

Normal probability plots were inspected to assess whether data were normally distributed or skewed. Normally distributed data were summarized as mean (standard deviation) while skewed data were summarized as median (range). For the percent sperm with an intact plasma membrane, the mean and variation were expressed as standard deviation (SD) for percent change from baseline and were analyzed using a mixed model two-way ANOVA. The model specified treatment (groups A–F), time (T1 for post-centrifugation, T2 for 24 h of cooling, T3 for 48 h of cooling), and the interaction between treatment and time as fixed effects. Dog identification was specified as a random effect. The interaction term was further analyzed to compare treatments within each time point and time points within each treatment. Recovery rate, subjective and computer assisted (CASA) motility, and sperm morphology variations were summarized as medians (range). Using the dog’s own baseline as his own control, the change from baseline (one outcome at a time) was analyzed using a linear generalized estimating equations model. The model specified treatment (groups A–F), time (T1–T3), and the interaction between treatment and time as fixed effects. Dog identification was included in the model as the subject of repetition. Repetition within the subject was modeled using the compound symmetry matrix specification. The interaction term was further analyzed to compare treatments within each time point and time points within each treatment. For all models, P values were adjusted for multiple comparisons using Tukey–Kramer’s procedure. Statistical analyses were performed with procedures available in SAS version 9.4 (Cary, NC, USA). Values were considered statistically significant when *p* < 0.05.

## 3. Results

The initial semen evaluation parameters of our study participants as a group are presented in Table 2.

### 3.1. Effect of Centrifugation Treatment on Spermatozoa Recovery Rate

As shown in Figure 1, sperm recovery rates calculated as RR1 and RR2 were similar, and there were no statistically significant differences by either method when comparing the treatment groups (A–F) (*p* > 0.9 and *p* > 0.06, respectively). The median recovery rate of all the treatment groups ranged from 99.2 to 102.1% for RR1 and 98.8 to 99.4% for RR2.

### 3.2. Effect of Centrifugation Treatment and Cooling on Sperm Plasma Membrane Integrity

There were no significant differences in spermatozoa plasma membrane integrity in terms of change from baseline between treatment groups at any of the time points analyzed (T1–T3) (*p* ≥ 0.38). On the other hand, cooling resulted in a similar decline in sperm membrane integrity in all six treatment groups from time points T1 to T2 (*p* ≤ 0.001) and T1 to T3 (*p* < 0.0001) (Figure 2). There was no difference when comparing the change in membrane integrity from T2 to T3 in any of the treatment groups (*p* ≥ 0.8) (Figure 2).

### 3.3. Effect of Centrifugation Treatment and Cooling on Sperm Motility

The change from baseline in subjective TM was similar across the six centrifugation treatment groups at all time points (T1–T3, *p* > 0.2). Over time, a statistically significant decline in subjective TM was found in all treatment groups for time points T1 to T2 (*p* ≤ 0.04), T1 to T3 (*p* ≤ 0.0002) and T2 to T3 (*p* ≤ 0.02) (Figure 3).

The treatment groups did not show statistically significant differences in TM (CASA) when groups were compared at each time point (T1–T3, *p* > 0.3). In contrast, all six treatment groups had a significant but similar decline in CASA TM when comparing time points T1 to T3 (*p* ≤ 0.02) and T2 to T3 (*p* ≤ 0.03) (Figure 4). There was no significant change in CASA TM for any of the groups between time points T1 and T2 (*p* > 0.7).

For the CASA PM evaluations, there were no significant differences in change from baseline between treatment groups at any time point (*p* > 0.4), except for treatment A compared to F at time point T3 (*p* ≤ 0.01). Cooling significantly decreased PM in all treatment groups from T1 to T3 (*p* ≤ 0.004), and from T2 to T3 (*p* ≤ 0.01) except for treatment group B. A significant decline between T1 and T2 (*p* ≤ 0.04) was found only for groups C and E (Figure 5).

### 3.4. Effect of Centrifugation Treatment and Cooling on Sperm Morphology 

In assessing morphology changes, abnormalities were grouped by region of the spermatozoa, i.e., acrosome, head, midpiece, and tail abnormalities. The change from baseline in the percentage of morphologically normal spermatozoa was not different between treatment groups at any time point (T1–T3; *p* ≥ 0.07) (Figure 6). However, there was a significant decline over time during the cooling period for all six treatment groups between time points T1 and T2 (*p* ≤ 0.0001) and T1 to T3 (*p* ≤ 0.0001), while no further decline was noted between T2 and T3 (*p* ≥ 0.9) (Figure 6).

For all changes regarding morphological abnormalities per treatment group at each time point, please see Appendix A. Overall, acrosomal defects ranged from 1.5 to 4.3% at T1, 9.5 to 16.3% at T2 and 17.8 to 20.8% at T3 for all six treatment groups. The percent change from baseline in acrosome abnormalities (i.e., knobbed, folded, wrinkled, swollen, detaching and detached acrosomes) was not significantly different across treatment groups at any of the time points (*p* > 0.15). As for time-related changes, a significant increase in acrosome abnormalities in all six treatment groups was noted from T1 to T2 and T1 to T3 (*p* < 0.0001), and from T2 to T3 for treatment groups A and B (*p* < 0.0001 and *p* = 0.02, respectively).

For head abnormalities, the highest percentage change was noted for treatment group B for T1, treatment groups B and C for T2, and treatment group B for T3. The lowest percentage change was noted for treatment group A at T1 and T2 and treatment group C at T3. There was no difference between treatment groups at any of the time points for head abnormalities (*p* ≥ 0.51). Time-related changes for head abnormalities were only noted for treatment groups B and D between T1 and T3 (*p* ≤ 0.02). In terms of percentage change from baseline for midpiece abnormalities, the only difference between treatment groups was found at T3 for groups D and F (*p* = 0.01). There was a difference for treatment group A between times T1 and T3 (*p =* 0.03) and for groups C and F between times T2 and T3 (*p =* 0.04 and *p =* 0.002, respectively). All other treatments and time point comparisons were not statistically significant (*p* ≥ 0.10). For tail abnormalities, only treatment groups C vs. E, and C vs. A at T2 differed (*p* = 0.002 and *p* = 0.04); there were no time related changes within any of the groups (*p* ≥ 0.07).

As we evaluated individual ejaculate aliquots and subjected them to the different centrifugation treatments, we noticed minor to moderate differences in the individual dog’s sperm membrane integrity, motility, and morphology parameter changes in response to centrifugation and cooling, however, these were not consistent across all animals, and a trend could not be identified.

## 4. Discussion

Centrifugation is a critical component in various assisted reproductive techniques such as semen freezing and processing for cooled shipment. The removal of prostatic fluid and seminal plasma from a canine ejaculate via centrifugation is commonly performed in clinical and experimental settings [24]. This practice is common across clinical andrology practices in other species as a first step in semen processing [10,25,26,27]. Studies have shown that equine, bovine, and ovine spermatozoa can be more tolerant of centrifugation in contrast to human or rodent sperm [16,19,20,25,27,28,29]. For example, stallion semen can be centrifuged at 1800–2400 g for 5 min with mild detrimental effects on semen quality parameters [13,20,30,31]. However, most of these semen samples are diluted with an extender prior to processing, which does not hold true for the routine processing of canine ejaculates [13]. Clinically, stallion semen is processed after extending in a 1:1 *vol:vol* ratio and centrifuged at a range between 400 g and 900 g for 9 to 15 minutes, depending on the stallion and the clinician’s preference [32]. Canine semen is not as tolerant as stallion’s to the higher centrifugation speeds of 1620 g or 2880 g as these resulted in compromised sperm membrane integrity after 48 and 72 h of cooled storage, yet 180 g led to higher sperm losses, leaving 720 g for 5 min as an appropriate setting for dogs [10]. Our work aimed at testing a narrower range of gravitational (g) forces and times of 5 and 10 min to find a range of acceptable settings, using individual ejaculates without pre-dilution. Using the raw ejaculate prior to admixture with an appropriate semen extender mimics the routine clinical processing in dogs [33,34]. Additionally, the cooling of the extended samples in a dedicated canine semen shipping box reflects the clinical aspect of shipping for artificial insemination use compared to cooling in a refrigerator, which is used in most research settings.

Our study showed that all centrifugation treatments were able to achieve complete sperm pellet formation. Sperm recovery rates were high (>98%, median) for both RR1 and RR2 without any significant differences between treatment groups, which highlights that processing will not significantly affect recovery rates for this range in gravitational (g) force and time. The RR1 values of over 100% may be due to the coefficient of variation (repeatability) of the Nucleocounter^®^ SP-100™, which is reported to be less than 4% for sperm concentration (total count) measurements, or to minor pipetting errors. However, hematocytometers were also used to calculate the recovery rate from the removed supernatant (RR2), which gave similar results to RR1, confirming the proper sample handling technique. Compared to previous work evaluating centrifugation settings for canine sperm by Rijsselaere et al. [10], our treatment groups (A–F) were in a similar range for sperm losses (−2.1 to 1.2% for both RR1 and RR2) to that study’s 720 g and 1620 g groups, where spermatozoa losses were noted to be 2.3% and 0.4%, respectively [10]. Our results, therefore, support that centrifuging canine raw ejaculates between 400 g and 900 g for 5–10 min is appropriate in terms of sperm losses, which are minimal and clinically irrelevant.

Additionally, we evaluated the effect of the different centrifugation treatment groups on several in vitro sperm parameters (membrane integrity, motility, morphology). These parameters were chosen as commonly used during clinical as well as research settings to estimate sperm fertilizing ability [33]. While there are in vitro and in vivo models to assess specific aspects of spermatozoa fertilizing ability in other species, we do not have such a capability in the dog [16,27,35]. The breeding soundness evaluation guidelines as established by the Society for Theriogenology and the American College of Theriogenologists correlate with the fertilizing ability of stud dogs. These guidelines previously established ≥70% total motility and ≥60% normal morphology to be considered the minimum to categorize a male as a satisfactory breeder [36]. However, as noted in bull andrology work, we need to understand the limitations of benchtop parameters for assessing the true fertilizing ability of males [16,27,37]. One consideration in our study is the use of dogs with a lower percentage of normal morphology due to the clinical population in our facility. In our clinic, the dogs with 40–50% normal morphology have shown normal fertility and are proven stud dogs, hence their inclusion in our study. However, this phenomenon of decreasing the percentage of morphologically normal semen with normal pregnancy rates and litter sizes is likely common in other clinics as canine breeders do not select for fertility.

Spermatozoa plasma membrane integrity demonstrated that no treatment was superior at any time point evaluated. The groups showed similar declines in the percentage of membrane intact sperm from baseline (T0) over time. While centrifugation treatment had no effect, cooling significantly decreased sperm membrane integrity during the first 24 h without a further drop at 48 h (T3). In the study by Rijsselaere et al. [10], lower centrifugation forces, of 180 g and 720 g, did not significantly negatively affect sperm membrane integrity compared to higher speeds, as there were no significant changes from 24 to 72 h of cooling. That study did not evaluate the treatment effect on the change in membrane intact spermatozoa immediately post-centrifugation to 24 h after cooling. For our study, we noted that this time frame had the most substantial drop in plasma membrane integrity (from T1 to T2), albeit without differences between centrifugation treatment groups. Accordingly, previous studies have shown that the length of time in cooled storage will negatively impact sperm plasma membrane integrity due to toxic effects from moribund spermatozoa and oxidative stress during storage [3,7].

For both subjective and computer assisted (CASA) sperm motility evaluations, centrifugation treatment groups were not significantly different from each other at any time point, except for treatment group A compared to F at T3 for CASA PM. The clinical relevance of that is likely minimal if any. Similar to membrane integrity, motility parameters also decreased over time, with subjective TM showing a more consistent negative trend across all time points, while CASA TM and PM essentially showed a significant decline by T3 only. Our finding agrees with the study of Rijsselaere et al., which noted a significant decline in TM and PM over the 72 h of cooling period independent of the centrifugation groups [10]. Other canine studies also reported negative effects of 24–96 h of cold storage on sperm motility variables [38,39]. This finding is similar to studies in other species, mainly stallions, where PM declines from initial to 48–72 h of cooled storage [20,40].

The percentage of morphologically normal spermatozoa declined significantly from the initial evaluation to 24 h of cooling without further a drop by 48 h, independent of the centrifugation treatment groups. Acrosomal defects were the most abundant among sperm abnormalities. Typically, acrosomal abnormalities are a large portion of the damage that progresses during cryopreservation or cooled storage of canine semen [41,42]. This is due to the glycoprotein and glycolipid changes as well as ROS production that can induce early onset capacitation in an in vitro setting during cold storage [8,19,43]. These acrosome abnormalities, while not specific to a certain centrifugation treatment group, are more indicative of the decline of plasma membrane integrity during the cooling process over time. The acrosomal changes we saw (swelling, wrinkling, detaching acrosomes, etc.) are consistent with previous work on canine semen, showing acrosomes to be the most affected during cooled storage or post-thaw [41,44,45]. Considering the importance of acrosome integrity as an in vitro measure of fertilization ability, other methods such as flow cytometry would provide an objective assessment of acrosome changes [46,47].

Our study did not adjust for a constant sperm concentration across samples from different dogs during cooling, which could have affected semen quality over time. However, the optimal sperm concentration for cooled storage of canine sperm is unknown, and clinically a *vol:vol* extension is currently used. However, this experimental design did not affect our ability to compare the different centrifugation settings as the same concentration was used across all treatment groups for each individual dog, allowing for treatment group comparisons at each time point. Furthermore, we used the change from baseline for all statistical evaluations, which adjusted for individual ejaculate differences, including concentration. By using the ejaculate of each dog instead of pooled semen samples, we were able to recognize individual differences in sperm parameters across centrifugation groups and over time. It is known that several factors affect the cooling ability of canine spermatozoa with some individuals performing better in a particular semen extender, or some dogs being more tolerant of the cooling process [11,12]. This may also be true for centrifugation settings, e.g., a dog performing better at 400 g over 720 g or 900 g, or vice versa. A limitation of our study is that we used otherwise healthy males with no known concerns for subfertility; therefore, subfertile males may have worse spermiogram results using these same centrifugation settings compared to our study population.

## 5. Conclusions

Based on these findings, our hypothesis is void as the centrifugation treatment groups were not significantly different in terms of sperm recovery rate and semen quality parameters when directly compared at each time point. Our conclusion is that centrifugation between 400 g and 900 g for 5 to 10 min is acceptable for processing raw canine semen when the removal of prostatic fluid and seminal plasma is necessary. Sperm losses are minimal and clinically irrelevant in these settings. Spermatozoa plasma membrane integrity, total and progressive motility, percentage of morphologically normal spermatozoa, and spermatozoa with acrosomal damage were also not different between these centrifugation settings. The decline in the percentage membrane intact and morphologically normal spermatozoa was more affected by post-centrifugation to 24 h of cooled storage for all treatment groups, while a significant decline in sperm motility presented from 24 h to 48 h during cooled storage. This implies that for best results, insemination with cooled canine semen should be performed no later than 24 h after semen collection and initial processing. For dogs with lesser quality semen, testing not only different extenders but different centrifugation gravitational (g) forces and times for optimal results may be important before shipping semen for insemination.

## Figures and Tables

**Figure 1 animals-13-01421-f001:**
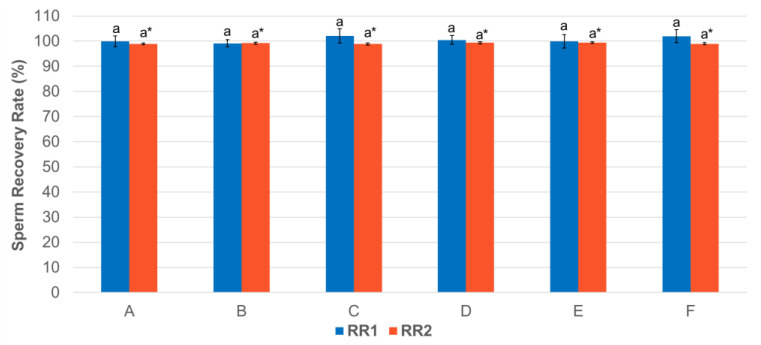
Spermatozoa recovery rate (RR, %) by centrifugation treatment groups. RR1 (blue columns) was calculated for each aliquot as (sperm concentration post-centrifugation and resuspension)/(sperm concentration in raw semen) × 100, using the Nucleocounter*^®^* (n = 14). RR2 (red columns) was calculated as (sperm concentration in raw semen*—*sperm concentration in supernatant)/(sperm concentration in raw semen) × 100, where the supernatant spermatozoa concentration was determined with a hemocytometer (n = 11). Columns denote the median and error bars denote the standard error of the mean (SEM). Treatment groups labeled as A: 400 g 5 min, B: 720 g 5 min, C: 900 g 5 min, D: 400 g 10 min, E: 720 g 10 min, F: 900 g 10 min. Columns with the same superscript across RR1 (a) or RR2 (a*) do not significantly differ (*p* > 0.05).

**Figure 2 animals-13-01421-f002:**
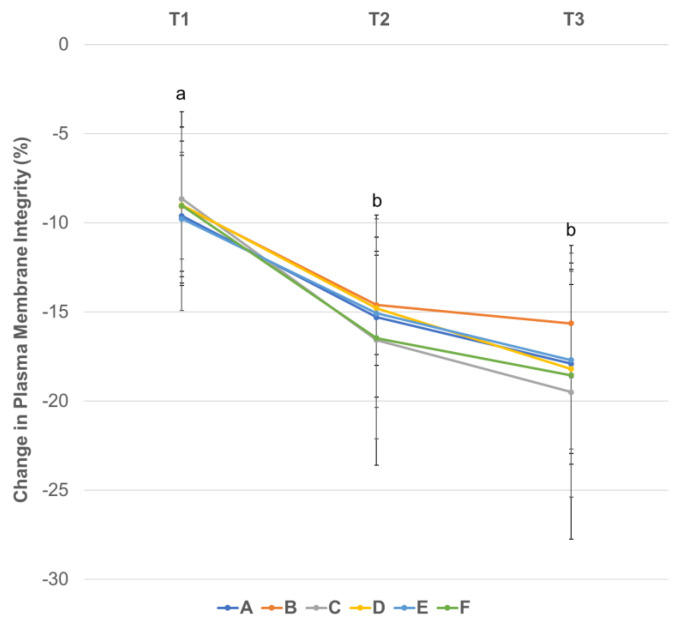
Change in spermatozoa plasma membrane integrity (%) from baseline (T0 for initial raw semen evaluation) by centrifugation treatment groups over time. Colored circles denote the mean and error bars denote the standard deviation (SD) per treatment group. Treatment groups labelled as A: 400 g 5 min, B: 720 g 5 min, C: 900 g 5 min, D: 400 g 10 min, E: 720 g 10 min, F: 900 g 10 min. Time points T1: post-centrifugation, T2: 24 h of cooling, T3: 48 h of cooling. Different letters (a, b) denote statistically significant (*p* < 0.05) difference between time points for all treatment groups.

**Figure 3 animals-13-01421-f003:**
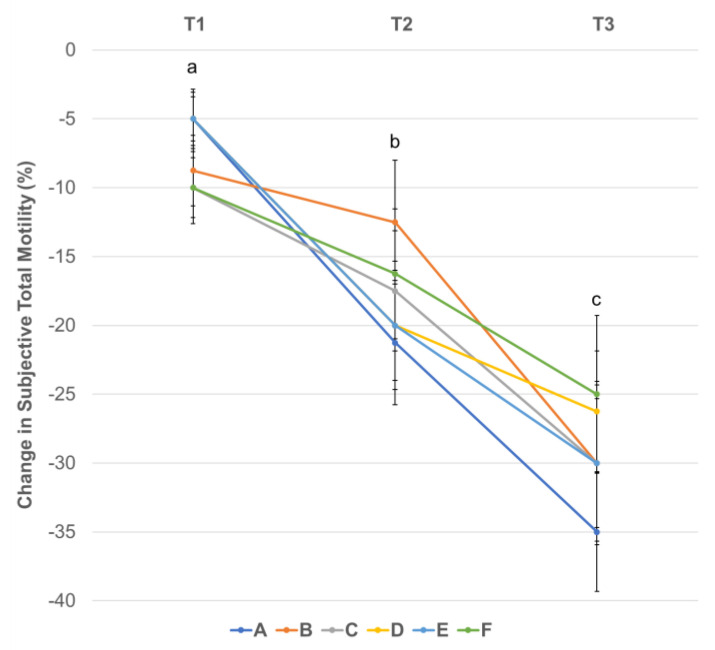
Change in subjective total motility (%) from baseline (T0 for initial raw semen evaluation) by centrifugation treatment groups over time during 48 h of cooling. Colored circles denote the median and error bars denote the standard error of the mean (SEM) per treatment group. Treatment groups labeled as A: 400 g 5 min, B: 720 g 5 min, C: 900 g 5 min, D: 400 g 10 min, E: 720 g 10 min, F: 900 g 10 min. Time points T1: post-centrifugation, T2: 24 h of cooling, T3: 48 h of cooling. Different letters (a, b, c) denote statistically significant (*p* < 0.05) difference between time points for all treatment groups.

**Figure 4 animals-13-01421-f004:**
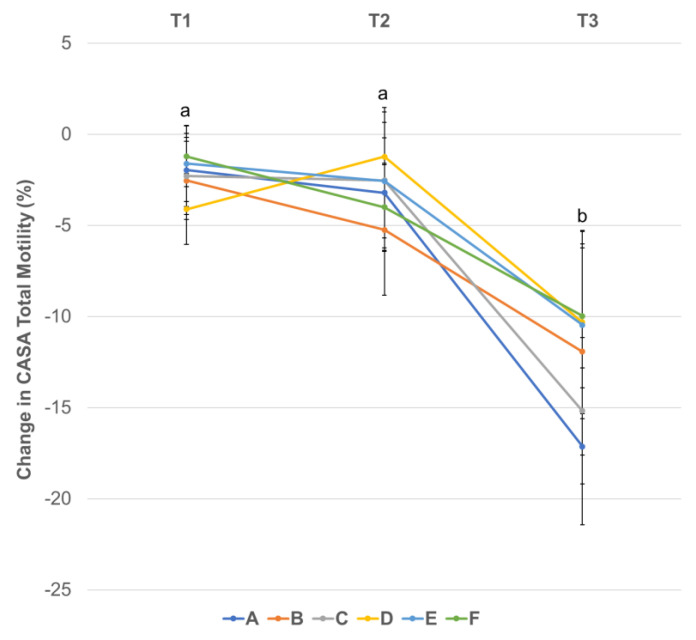
Change in total motility (%) analyzed by CASA from baseline (T0 for initial raw semen evaluation) by centrifugation treatment groups over time during 48 h of cooling. Colored circles denote the median and error bars denote the standard error of the mean (SEM) per treatment group. Treatment groups labeled as A: 400 g 5 min, B: 720 g 5 min, C: 900 g 5 min, D: 400 g 10 min, E: 720 g 10 min, F: 900 g 10 min. Time points T1: post-centrifugation, T2: 24 h of cooling, T3: 48 h of cooling. Different letters (a, b) denote statistically significant (*p* < 0.05) differences between time points for all treatment groups.

**Figure 5 animals-13-01421-f005:**
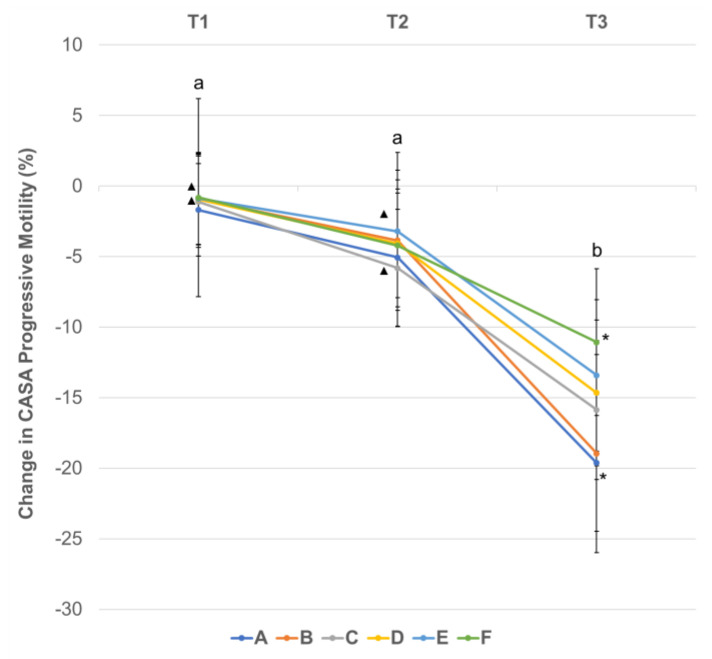
Change in progressive motility (%) analyzed by CASA from baseline (T0 for initial raw semen evaluation) by centrifugation treatment groups over time during 48 h of cooling. Colored circles denote the median and error bars denote the standard error of the mean (SEM) per treatment group. Treatment groups labeled as A: 400 g 5 min, B: 720 g 5 min, C: 900 g 5 min, D: 400 g 10 min, E: 720 g 10 min, F: 900 g 10 min. Time points T1: post-centrifugation, T2: 24 h of cooling, T3: 48 h of cooling. Different letters (a, b) denote statistically significant (*p* < 0.05) differences between time points for all treatment groups, except for group B between T2 and T3. Black triangles (▲) denote statistically significant (*p* < 0.05) differences between time points for the given groups. Asterisks (*) denote statistically significant (*p* < 0.05) differences between treatment groups at a given time point.

**Figure 6 animals-13-01421-f006:**
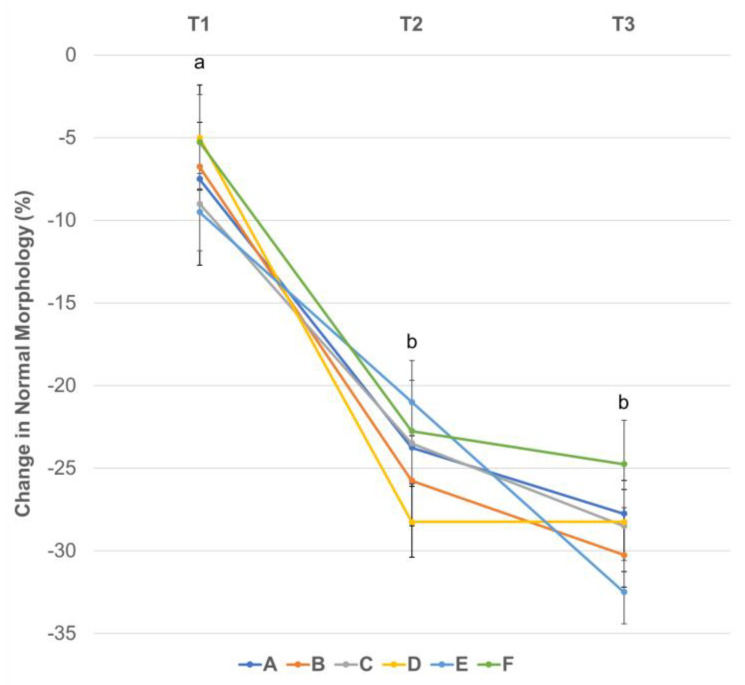
Change in morphologically normal spermatozoa (%) from baseline (T0 for initial raw semen evaluation) over time by centrifugation treatment groups. Colored circles denote the median and error bars denote the standard error of the mean (SEM) per treatment group. Treatment groups labeled as A: 400 g 5 min, B: 720 g 5 min, C: 900 g 5 min, D: 400 g 10 min, E: 720 g 10 min, F: 900 g 10 min. Time points T1: post-centrifugation, T2: 24 h of cooling, T3: 48 of cooling. Different letters (a, b) denote statistically significant (*p* < 0.05) differences between time points for all treatment groups.

**Table 1 animals-13-01421-t001:** Technical settings of the Computer Assisted Sperm Analysis (CASA) system (Sperm Vision™, Minitube).

Parameter	Setting
Field-of-view depth (sample chamber depth)	20 µm
Light adjustment	80–110
Total number of fields	7 fields
Sperm recognition area	20–60 µm^2^
Frame rate	60 frames/s
Points assessed for sperm motility	11
Total motility	Progressive motility + local motility
Immotile sperm	AOC* < 9.5
Progressive motility	Every cell that is not “immotile” or “local motile”

*AOC: average orientation change of head.

**Table 2 animals-13-01421-t002:** Initial semen evaluation results of the fresh (raw) ejaculates for the whole study population (n = 14).

Parameter	Mean ± S.D.
Volume (mL)	4.25 ± 1.83
Concentration (×10^6^/mL)	175.33 ± 63.08
Viability (%)	87.1 ± 6.57
Subjective Total Motility (%)	83.75 ± 6.85
CASA Total Motility (%)	86.91 ± 5.05
CASA Progressive Motility (%)	81.77 ± 11.30
Morphologically Normal Spermatozoa (%)	51.32 ± 9.52
Acrosome defects (%)	3.11 ± 2.25
Head defects (%)	23.86 ± 7.95
Midpiece defects (%)	16.71 ± 5.17
Tail defects (%)	9.25 ± 5.64

## Data Availability

The data presented in this study belong to the authors and are available on reasonable request from the corresponding author.

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
