# Peer review of "Defining an Optimal Range of Centrifugation Parameters for Canine Semen Processing"

_animals, 2023, doi:10.3390/ani13081421_

Round 1

Reviewer 1 Report

In this study, Nicole Sugai et al. explored the effect of centrifugation g force (400, 720, and 900g) and time (5 and 10 min) on canine sperm recovery rate, motility (total and progressive), and morphology during cool storage. There were no differences between treatments in any parameter, except for a significantly higher progressive motility in samples exposed to 900g for 10 min compared to those centrifuged at 400 g for 5 min. The manuscript provides some insights into optimal conditions for seminal centrifugation, but there are major issues that need to be addressed.

Major comments

-The authors used a CASA system but only evaluated the sperm total and progressive motility. I recommend including the kinematic parameters that are assessed by this objective instrument (i.e., VAP, VCL, VSL, ALH, BCF, STR and LIN).

-Sperm concentration should have been standardized for sperm motility analysis as it may affect the results. This flaw must be discussed as a major limitation of the study.

-Acronyms’ explanation is missing in Table 1. Acronyms explanation is especially useful to clarify the definition of total and progressive motility. Hyperactive and other sperm subpopulations should be removed unless the authors include these results in the manuscript. Total frames acquired or video duration must be included.

-It is not clear the rationale behind the use of a Leja chamber for CASA and a microscope slide with coverslip for assessing the subjective motility. Why did not the authors use the Leja chamber for both analyses? Besides, microscope slides and coverslips are not recommended for motility assessment as the chamber depth is not standardized. I would consider removing the subjective sperm motility assessment from the manuscript.

-The authors need to explain why they used the Nucleocell counter for sperm recovery rate in the sediment and hemocytometer in the supernatant.

-Please include further information about the cooling process. How were the samples cooled? Which was the cooling rate?

-In line 392 the authors state that ≥60% normal sperm are required to classify a dog as a satisfactory breeder, but the samples used in this study showed lower percentage of normal sperm (i.e., 51.32±9.52, Table 2). How do the authors explain such a low sperm morphology? Please, include the explanation in the manuscript.

-Comparisons are made based on percentage of change compared to the baseline value of raw semen. I recommend including the comparison of means between raw semen (T0) and post-centrifugation (T1) and cooled (T2 and T3) values per each parameter.

-Line 424-428 Provide a clearer explanation. Based on your statement, a higher percentage of acrosomal defects because of agglutination should be expected in treatments with higher g force and time instead of the treatments exposed to 400 and 720 g for 5 min.

Minor comments

Line 161 Does “1000x” refer to magnification or objective? Please replace the letter x with the symbol ×. As sperm morphology was evaluated by 2 observers, were the results averaged?

Lines 334-336 Please, indicate which treatment showed the highest and which the lowest percentage of head abnormalities.

Author Response

Please see the attachment for full explanation. 

Reviewer 2 Report

This reviewed manuscript describes a study which aimed to determine the optimal range of centrifugation parameters needed for cooled canine semen from 14 healthy male dogs. Dogs were split into 6 treatment groups and used as their own control. Semen from these dogs was processed and evaluated in a clinically relevant way, which yielded the results that 400g-900g for 5-10 mins was the appropriate processing speed and time.

Overall, this was an elegant study that was well designed. Sample size appeared adequate and the fact that the samples were processed and evaluated in a way that would be performed practice was very helpful. The suggestions I have are minor as the authors did a very thorough job with this study design and execution. The brief comments I have are as follows:

- Lines 60-68: References the Rijsselaere study of which I was already familiar. It may be helpful to include a brief sentence here on how your study differed in design compared to this study

- Line 70-71: "...those studies did not specifically investigate the effects of particular centrifugation parameters on semen quality." - In what way? How was your study different? A simple short explanation should be sufficient here.

-Line 70: Typo in Rijsselaere 

Section 2.2, Line 124: Consider "manual technique" instead of digital maniplation - to me, digital manipulation means you just pinched it or poked it with your finger...

Line 133-134: How were these cut-offs for total motility and morphologically determined? Why not progressive motility?

References:

- Reference 3 - typo "stored" is written "sotred"

- Double check formatting. Some journals are abbreviated while others aren't (this may be ok?)

- Weird spacing on references 24 and 36 (check justification)

- Reference 31: 2nd edition is written twice "Second edition 2nd ed"

Author Response

This reviewed manuscript describes a study which aimed to determine the optimal range of centrifugation parameters needed for cooled canine semen from 14 healthy male dogs. Dogs were split into 6 treatment groups and used as their own control. Semen from these dogs was processed and evaluated in a clinically relevant way, which yielded the results that 400g-900g for 5-10 mins was the appropriate processing speed and time.

Overall, this was an elegant study that was well designed. Sample size appeared adequate and the fact that the samples were processed and evaluated in a way that would be performed practice was very helpful. The suggestions I have are minor as the authors did a very thorough job with this study design and execution. The brief comments I have are as follows:

Thank you very much for the review and your time to evaluate our study. We will take the comments to help with the revision of the manuscript and to improve the quality of our manuscript.

- Lines 60-68: References the Rijsselaere study of which I was already familiar. It may be helpful to include a brief sentence here on how your study differed in design compared to this study

For this reference, we edited and added an additional explanation for comparison with the work of Rijsselaere et al. We also discussed the difference in our study design compared to their study design in lines 362-373 (updated to lines 432-438). We can add more detail for our study design compared to other studies as mentioned in this comment and the next.

- Line 70-71: "...those studies did not specifically investigate the effects of particular centrifugation parameters on semen quality." - In what way? How was your study different? A simple short explanation should be sufficient here.

Thank you for this question. We revised this section to explain why we mentioned these studies.

-Line 70: Typo in Rijsselaere 

Thank you for catching that mistake. It is fixed now.

Section 2.2, Line 124: Consider "manual technique" instead of digital maniplation - to me, digital manipulation means you just pinched it or poked it with your finger...

Thank you for the comment. We adjusted the wording accordingly.

Line 133-134: How were these cut-offs for total motility and morphologically determined? Why not progressive motility?

Thank you for this comment and question. Due to our clinical practice, we use the total subjective motility for calculating total normal motile sperm and breeding doses, and we evaluate progressive motility on a 0-5 score. With that said, based on Hernandez-Aviles et al 2021 (Cool-stored and frozen-thawed stallion semen: thoughts on collection, evaluation, processing, insemination, and fertility) comments about motility, there is bias to progressive motility. The quality of sperm motion can vary and may not completely be associated with pathology or infertility for the male. For our initial semen parameters, we elected to use the established cutoffs from the SFT guidelines (Purswell et al 2015) for total motility (≥70%). For the percentage normal morphology, our original goal was ≥60%. However our clinical contingent has trended downward, and we are seeing 40-50% normal routinely. Hence our adjustment to the ≥40% for inclusion in the study. We added a statement in the manuscript as well to help explain this. See lines 509 to 520.  “One consideration in our study is the use of dogs with lower percentage normal morphology due to the clinical population in our facility. In our clinic, the dogs with 40-50% normal morphology have shown normal fertility and are proven stud dogs, hence their inclusion to our study. However, this phenomenon of decreasing percentage of morphologically normal semen with normal pregnancy rates and litter sizes is likely common in other clinics as canine breeders do not select for fertility.” 

References:

- Reference 3 - typo "stored" is written "sotred"

Thank you for that catch. This should be fixed from the endnote library

- Double check formatting. Some journals are abbreviated while others aren't (this may be ok?)

Thank you for that catch. This should be fixed from the endnote library.

- Weird spacing on references 24 and 36 (check justification)

Thank you for that catch. I believe the spacing is fixed for this section to be more uniform.

- Reference 31: 2nd edition is written twice "Second edition 2nd ed"

Thank you for that catch. This should be fixed from the endnote library

Reviewer 3 Report

Although the scientific level of the publication is average, it should be taken into account that the experiment was very well prepared and conducted, and the results obtained will be useful in further research on the use of dog semen for insemination. The obtained results may also be useful in research on insemination of other animal species.

Author Response

Thank you very much for your comments and review of the manuscript.

Round 2

Reviewer 1 Report

The authors addressed all my comments.